# Association of bariatric surgery with all-cause mortality and incidence of obesity-related disease at a population level: A systematic review and meta-analysis

Tom Wiggins[1,2], Nadia Guidozzi[1], Richard Welbourn[2], Ahmed R. Ahmed[1], Sheraz R. Markar[1,3]*

1 Department of Surgery and Cancer, Imperial College London, London, United Kingdom, 2 Department of Bariatric Surgery, Musgrove Park Hospital, Taunton, United Kingdom, 3 Department of Molecular Medicine & Surgery, Karolinska Institutet, Stockholm, Sweden

* s.markar@imperial.ac.uk

## Abstract

**Data Availability Statement:** All relevant data are available within the manuscript and its Supporting information files.

### Background

Previous clinical trials and institutional studies have demonstrated that surgery for the treatment of obesity (termed bariatric or metabolic surgery) reduces all-cause mortality and the development of obesity-related diseases such as type 2 diabetes mellitus (T2DM), hypertension, and dyslipidaemia. The current study analysed large-scale population studies to assess the association of bariatric surgery with long-term mortality and incidence of new-onset obesity-related disease at a national level.

### Methods and findings

A systematic literature search of Medline (via PubMed), Embase, and Web of Science was performed. Articles were included if they were national or regional administrative database cohort studies reporting comparative risk of long-term mortality or incident obesity-related diseases for patients who have undergone any form of bariatric surgery compared with an appropriate control group with a minimum follow-up period of 18 months. Meta-analysis of hazard ratios (HRs) was performed for mortality risk, and pooled odds ratios (PORs) were calculated for discrete variables relating to incident disease. Eighteen studies were identified as suitable for inclusion. There were 1,539,904 patients included in the analysis, with 269,818 receiving bariatric surgery and 1,270,086 control patients. Bariatric surgery was associated with a reduced rate of all-cause mortality (POR 0.62, 95% CI 0.55 to 0.69, $p <$ 0.001) and cardiovascular mortality (POR 0.50, 95% CI 0.35 to 0.71, $p <$ 0.001). Bariatric surgery was strongly associated with reduced incidence of T2DM (POR 0.39, 95% CI 0.18 to 0.83, $p =$ 0.010), hypertension (POR 0.36, 95% CI 0.32 to 0.40, $p <$ 0.001), dyslipidaemia (POR 0.33, 95% CI 0.14 to 0.80, $p =$ 0.010), and ischemic heart disease (POR 0.46, 95% CI 0.29 to 0.73, $p =$ 0.001). Limitations of the study include that it was not possible to account for unmeasured variables, which may not have been equally distributed between patient

**Funding:** SRM - National Institute of Health Research (NIHR) Imperial Biomedical Research Centre grant (Grant number NIHR-IAT-EAN/021/00146/C). Sponsors played no role in study design, data collection or analysis, decision to publish or manuscript preparation.

**Competing interests:** The authors have declared that no competing interests exist.

**Abbreviations:** BMI, body mass index; HR, hazard ratio; OR, odds ratio; POR, pooled odds ratio; T2DM, type 2 diabetes mellitus.

groups given the non-randomised design of the studies included. There was also heterogeneity between studies in the nature of the control group utilised, and potential adverse outcomes related to bariatric surgery were not specifically examined due to a lack of available data.

## Conclusions

This pooled analysis suggests that bariatric surgery is associated with reduced long-term all-cause mortality and incidence of obesity-related disease in patients with obesity for the whole operated population. The results suggest that broader access to bariatric surgery for people with obesity may reduce the long-term sequelae of this disease and provide population-level benefits.

## Author summary

### Why was this study done?

- Surgical treatment for obesity (known as bariatric or metabolic surgery) has been suggested to reduce risk of death in people with obesity and decrease development of other medical conditions including diabetes, heart disease, and certain forms of cancer.
- This study aimed to analyse pooled data from previously published studies utilising data from national or regional administrative databases in order to examine the effect of bariatric surgery at a population level.

### What did the researchers do and find?

- This study pooled data from published research articles utilising large administrative databases (with information from over 1.5 million patients)
- The results suggested that bariatric surgery was associated with reduced risk of death, and decreased incidence of new-onset diabetes, high blood pressure, high cholesterol, and heart disease.

### What do these findings mean?

- These findings suggest that bariatric surgery is associated with reduced risk of premature death or developing other medical conditions in people with obesity at a population level.
- Increased availability of bariatric surgery may help to improve health outcomes for these individuals.

## Introduction

Obesity is a worldwide pandemic, with 39% of the worldwide adult population being considered overweight [1]. Mean worldwide body mass index (BMI) has been steadily increasing since 1975, and current trends predict that 20% of the global population will be classified as having obesity (BMI $\geq$ 30 kg/m$^2$) by 2030 [1,2]. In the UK, 27% of the population is already classified as having obesity [3]. Billions of dollars are spent annually worldwide to curb this health crisis, alongside the development of government public health initiatives [2,4].

Secondary to smoking, obesity has been identified as the leading cause of preventable premature death in the US [5]. Obesity predisposes individuals to a plethora of adverse conditions and is an established risk factor for overall all-cause mortality, cardiovascular disease, and type 2 diabetes mellitus (T2DM) [6–9]. Although multiple therapeutic options for the treatment of patients with obesity are available, bariatric surgery has been established as the most effective method of achieving sustained weight loss in these patients [10,11]. Bariatric surgery has also been associated with reduced overall mortality rates in patients with obesity [12] and leads to remission of obesity-related disease [10,11,13]. Bariatric surgery has also been associated with an overall reduction in the risk of developing multiple cancer types [14–16].

Current national guidance in the UK recommends that bariatric surgery be considered for patients who have a BMI of 40 kg/m$^2$ or more, or a BMI between 35 kg/m$^2$ and 40 kg/m$^2$ if other significant obesity-related comorbidities are present [17]. Patients with new-onset T2DM may also be considered for surgery at lower BMI ($\geq$30 kg/m$^2$). Despite the presence of these criteria and evidence that patients are healthier and more functional following bariatric surgery, far less than 1% of eligible patients receive bariatric surgery [18,19]. Data from the UK National Bariatric Surgery Registry establish that at the time of primary surgery 53.9% of men and 41.4% of women already have a high level of co-existing disease (defined as 4 or more obesity-related diseases) [20]. Previous studies have also demonstrated that improving pathways to bariatric surgery can reduce the healthcare burden upon individuals and lead to overall healthcare cost savings [21].

Results from previous pooled analyses have demonstrated that bariatric surgery reduces long-term all-cause mortality [22,23]. However, these studies also included data from clinical trials and single-institution studies, which may lack external validity compared to data collected independently in population studies undertaken using administrative datasets. The present study aimed to investigate, in people with severe and complex obesity, the influence of bariatric surgery on overall long-term mortality and prevention of incident obesity-related disease at a population level.

## Methods

A systematic literature search of Medline (via PubMed), Embase, and Web of Science was performed using the search criteria as follows: ("Bariatric Surgery"[MeSH] OR bariatric surgery) AND (mortality OR survival OR diabetes OR metabolic OR hypertension OR sleep apnoea OR cardiac OR angina OR heart disease OR myocardial infarction OR dyslipidaemia OR dyslipidemia OR thromboembolism) AND (national OR registry OR population) (S1 Text). Two authors (TW and NG) performed the electronic literature search independently, which was last updated in January 2020. The literature search dates included any studies published between 1 January 2000 and 31 January 2020. Studies published prior to 2000 were excluded to ensure all data were contemporaneous and applicable to present-day bariatric surgical practice. The electronic search was supplemented by a hand-search of published abstracts from relevant specialist conference meetings. Reference lists of all relevant studies were also reviewed

to identify potentially relevant studies. The protocol for this study was not prospectively registered in any repository for systematic reviews.

Identified abstracts were independently scrutinised by 2 reviewers (TW and NG) to determine eligibility for inclusion. Any discrepancies regarding study inclusion from the literature search were settled by discussion with a third author (SRM). Studies were included if they were either national or regional administrative database cohort studies reporting comparative risk of long-term mortality or incident obesity-related diseases for patients who have undergone any form of bariatric surgery compared to an appropriate control group (i.e., with a clinical diagnosis of obesity) with a minimum follow-up period of 18 months. For the purposes of this study, obesity-related diseases were defined as T2DM, hypertension, obstructive sleep apnoea, cardiac disease (ischemic heart disease or cardiac failure), dyslipidaemia, and venous thromboembolism. Studies that only provided details on remission of obesity-related diseases existing prior to bariatric surgery (as opposed to new onset) were not included. Any study that was not a population-based study utilising data from an administrative database (including randomised controlled trials and single-institution studies) was excluded. This selection criterion was imposed to ensure that results from all included studies could be directly applied at a population level. This led to randomised controlled studies, and cohort studies that did not utilise a population-registry-based data design, being excluded as results from these studies may be representative of only the highly specialised units participating in such studies and would lack external validity when applied to the general population of patients receiving bariatric surgery. In the situation where 2 studies utilised the same registry data and reported on the same outcome measure, and therefore potentially contained duplicate data, the most recent study was selected for inclusion. Only English-language studies were included.

Data from eligible studies were extracted into a computerised spreadsheet for analysis. Data were collected for overall all-cause long-term mortality and new-onset obesity-related comorbidities. Authors of individual studies were not specifically contacted to obtain more detailed results than those available in the main publication and published appendices. Mortality was evaluated according to reported HRs to minimise the effect of patient dropouts, whereas obesity-related diseases were primarily analysed directly according to event rates in each group. Event rate analysis was used for the primary analysis of obesity-related disease, as a greater proportion of included studies reported these data, but outcomes were also analysed according to adjusted odds ratios (ORs) where these data were available, for confirmatory purposes.

Statistical analysis was performed using StatsDirect 3.2.9. Pooled outcome measures were determined using random effects models as described by DerSimonian and Laird [24]. Heterogeneity amongst the trials was assessed by Cochran $Q$ statistic, a null hypothesis test in which $p < 0.05$ is taken to indicate the presence of heterogeneity, and the $I^2$ statistic, which describes the percentage of variation across studies due to heterogeneity. The Egger test was used to assess the funnel plot for asymmetry, indicating possible publication or other biases.

## Results

The literature search identified 18 studies suitable for inclusion [25–42]. Fig 1 provides details of the PRISMA flowchart for the literature search. In total there were 1,539,904 patients included in the analysis, with 269,818 patients receiving bariatric surgery and 1,270,086 control patients. The types of surgery were gastric bypass ($n$ = 137,578, 51%), sleeve gastrectomy ($n$ = 58,916, 22%), adjustable gastric band ($n$ = 52,973, 20%), vertical banded gastroplasty ($n$ = 6,397, 2%), biliopancreatic diversion (with or without duodenal switch) ($n$ = 1,002, 0.4%), and an alternative procedure or unspecified operation ($n$ = 12,952, 5%) (S1 Table). Median follow-up across all studies was 59 months (range 18 to 144 months).

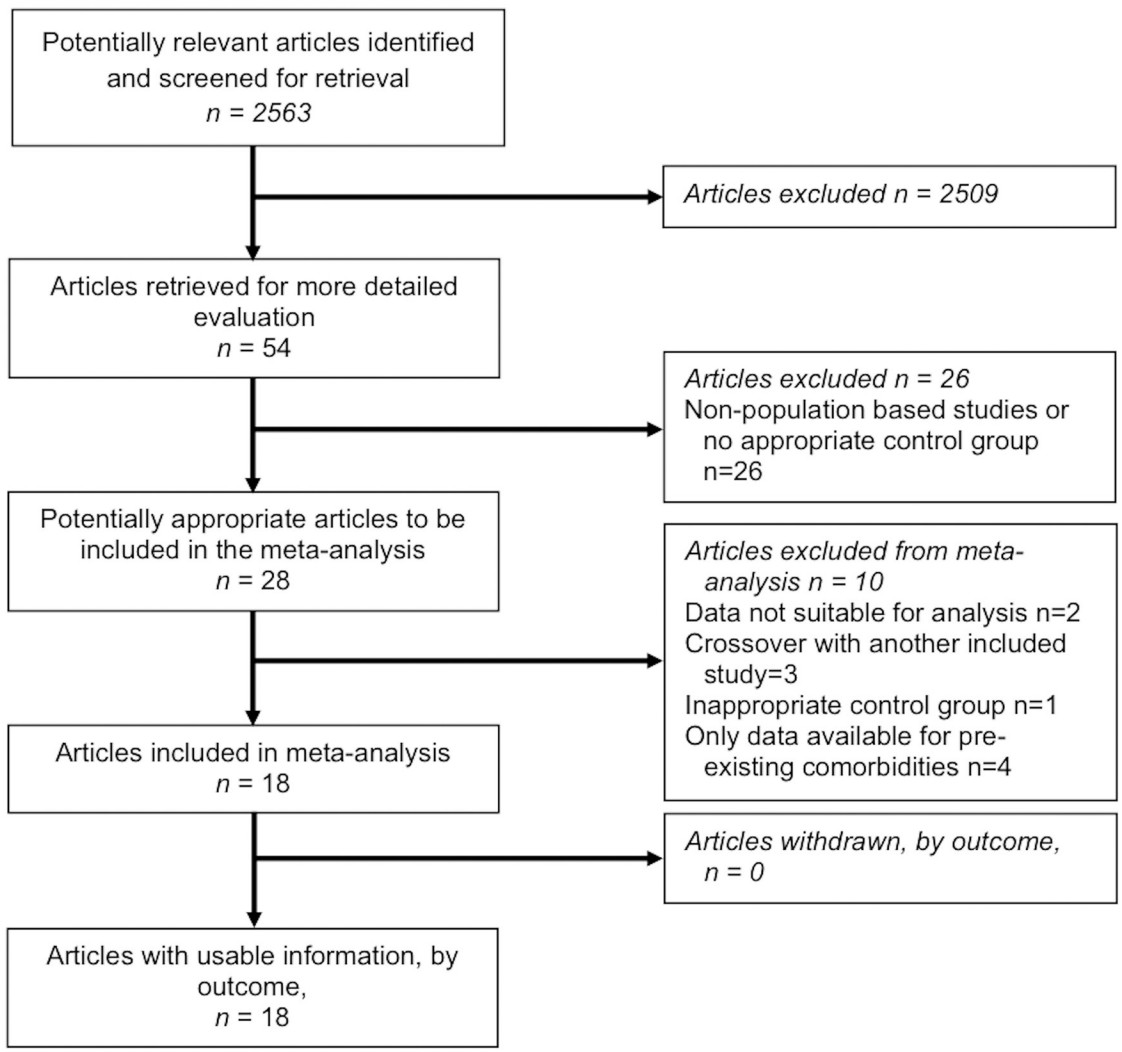

**Fig 1. PRISMA flow chart with details of literature search.**

Patient demographic details for all studies are provided in Table 1. Quality assessment of studies was undertaken with the Newcastle–Ottawa Scale (S2 Table) [43]. The majority of studies used patients with a diagnosis of obesity as the control group, although a group of patients without a diagnosis of obesity was utilised as the control group in 1 study [26]. All studies except this one scored 4 stars for patient selection (rated out of 4) [25,27–42]. Studies that did not report BMI for control and surgery groups scored only 1 star for comparability [26,28,29,39–42]. All studies scored the maximum 3 stars for exposure.

## All-cause mortality

Eleven studies reported a significant reduction in relative risk of long-term all-cause mortality for patients following bariatric surgery compared to controls [25,28,30–32,36,37,38,39,41,42], with a pooled OR (POR) of 0.62 (95% CI 0.55 to 0.69, $p < 0.001$). There was some evidence of statistical heterogeneity (Cochran $Q = 39.03$, $p < 0.001$; $I^2 = 71.8\%$) but no evidence of bias (Egger intercept = −0.31, $p = 0.714$) (Fig 2).

**Table 1. Patient demographics for the studies included in the pooled analysis.**

| Study | Nation or region of origin | Included study dates | Median follow-up (months) | Number of patients | | Mean ± SD or median (range) age (years) | | Female sex, *n* (%) | | Mean BMI ± SD (kg/m$^2$) or *n* (%) with BMI ≥ 40 kg/m$^2$ | |
|---|---|---|---|---|---|---|---|---|---|---|---|
| | | | | Surgery | Control | Surgery | Control | Surgery | Control | Surgery | Control |
| Arterburn [25] | US | 2000–2011 | 60 | 2,500 | 7,462 | 52 ± 8.8 | 53 ± 8.7 | 651 (26%) | 1,920 (26%) | 47 ± 7.9 | 46 ± 7.3 |
| Backman [26] | Sweden | 2007–2012 | 55 | 18,418 | 175,138 | 39 ± 10.5 | 39 ± 10.5 | 14,518 (79%) | 138,082 (79%) | 42.2 ± 5.8 | — |
| Bailly [35] | France | 2008–2016 | 44 | 102,627 | 225,882 | 37.3 ± 10.5 | 44.0 ± 11.7 | 88,464 (86%) | 151,566 (67%) | 47,516 (46.3%) | 37,948 (16.8%) |
| Ceriani [36] | Lombardy region, Italy | 1999–2008 | 144 | 472 | 1,405 | 43.1 ± 10.6 | 43.5 ± 12.5 | 354 (75%) | 995 (71%) | 47.3 ± 7.5 | 46.8 ± 3.8 |
| Douglas [37] | UK | Up to 2014 | 36 | 3,882 | 3,882 | 45 ± 11 | 45 ± 11 | 3,126 (81%) | 3,166 (82%) | 44.7 ± 8.8 | 42.1 ± 6.5 |
| Eliasson [38] | Sweden | 2007–2014 | 60 | 6,132 | 6,132 | 48.5 ± 9.8 | 50.5 ± 12.7 | 3,678 (61%) | 3,768 (61%) | 42.0 ± 5.7 | 41.4 ± 5.7 |
| Flum [39] | Washington State, US | 1987–2001 | 120 | 3,328 | 62,781 | 43.1 ± 10.1 | 47.0 ± 6.2 | 2,679 (81%) | 40,368 (64%) | — | — |
| Johnson [40] | South Carolina, US | 1996–2009 | 60 | 2,580 | 13,371 | 47.5 ± 10.6 | 52.1 ± 12.8 | 1,987 (77%) | 9,012 (67%) | — | — |
| Kauppila [41] | Nordic countries | 1980–2012 | 48 | 49,977 | 494,842 | — | — | 37,247 (75%) | 334,407 (68%) | — | — |
| Moussa [42] | UK | Up to 2017 | 129 | 3,842 | 177,973 | — | — | — | — | — | — |
| Moussa [27] | UK | Up to 2017 | 60 | 4,073 | 4,073 | 50 (42–58) | 50 (43–58) | — | — | 40.2 (37.0–45.2) | 40.4 (36.7–45.6) |
| Perry [28] | US | 2001–2014 | 18 | 11,903 | 11,901 | — | — | 9,237 (78%) | 9,236 (78%) | — | — |
| Persson [29] | Sweden | 1987 onwards | 44 | 22,295 | 25,564 | 40.7 ± 10.7 | 44.3 ± 13.2 | 16,921 (76%) | 17,077 (67%) | — | — |
| Pontiroli [30] | Lombardy region, Italy | 1995–2001 | 59 | 385 | 681 | 39.2 ± 10.4* | 40.2 ± 12.0* | 292 (76%) | 509 (75%) | 41.1 ± 5.4* | 40.9 ± 7.3* |
| Singh [31] | UK | 1990–2018 | 43 | 5,170 | 9,995 | — | — | 4,158 (80%) | 8,105 (81%) | 3,634 (70.3%) | 6,780 (67.8%) |
| Reges [32] | Israel | 2005–2014 | 48 | 8,385 | 25,155 | 46 (37–54) | 46 (37–54) | 5,490 (66%) | 16,470 (66%) | 4,980 (59%) | 14,940 (59%) |
| Thereaux [33] | France | 2008–2015 | 72 | 15,650 | 15,650 | 38.9 ± 11.2 | 39.4 ± 11.2 | 13,241 (85%) | 13,241 (85%) | 9,449 (60%) | 9,449 (60%) |
| Thereaux [34] | France | 2009 | 72 | 8,199 | 8,199 | 39.9 ± 11.5 | 40.5 ± 11.6 | 6,728 (82%) | 6,728 (82%) | 6,092 (74%) | 6,092 (74%) |

*Study subgroup with no type 2 diabetes at baseline.

## Cardiovascular mortality

Three studies reported significantly reduced relative risk of cardiovascular mortality for patients following bariatric surgery compared to controls [36,38,41] (POR 0.50, 95% CI 0.35 to 0.71, $p < 0.001$). There was no evidence of statistical heterogeneity (Cochran $Q = 2.82$, $p = 0.24$; $I^2 = 29.2\%$), and too few strata to assess for bias.

Data for overall and cardiovascular mortality are presented in Table 2.

## Details of incident comorbidities

Details of incident comorbidities in each study are provided in Table 3.

## Summary meta-analysis plot [random effects]

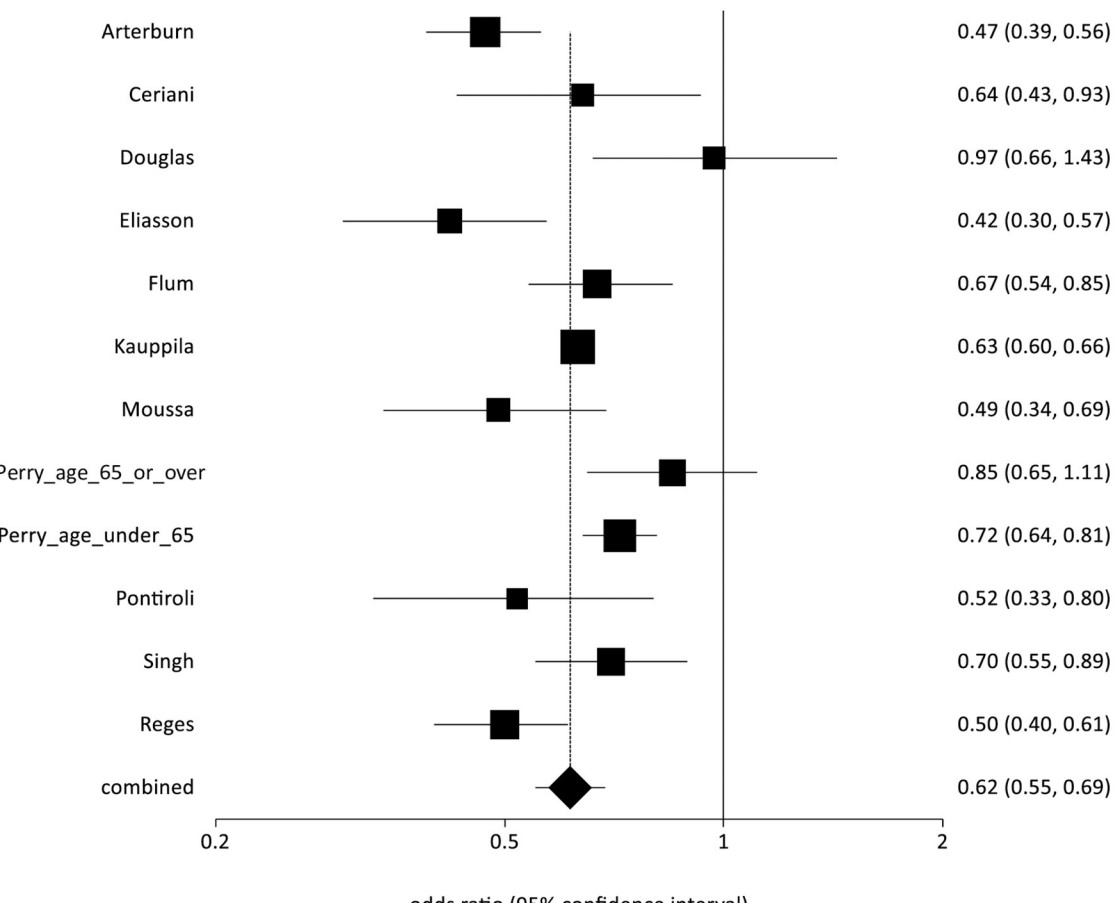

Fig 2. Forest plot of all-cause mortality (pooled odds ratio 0.62, 95% CI 0.55 to 0.69, $p < 0.001$).

### New-onset type 2 diabetes

Six studies reported a reduction in incident T2DM after bariatric surgery compared to controls [26,30,32,33,35,37] (POR 0.39, 95% CI 0.18 to 0.83, $p = 0.010$) (Fig 3). There was significant evidence of statistical heterogeneity (Cochran $Q = 824.8$, $p < 0.001$; $I^2 = 99.4\%$) but no evidence of bias (Egger intercept = 8.18, $p = 0.069$).

### New-onset hypertension

Five studies reported that incident hypertension was reduced after bariatric surgery compared to controls [30–32,34,37] (POR 0.36, 95% CI 0.32 to 0.40, $p < 0.001$) (Fig 4). There was no evidence of statistical heterogeneity (Cochran $Q = 5.90$, $p = 0.21$; $I^2 = 32.2\%$) and no evidence of bias (Egger intercept = −0.32, $p = 0.92$).

### New-onset obstructive sleep apnoea

Only 1 study reported incident obstructive sleep apnoea relative to controls, and therefore it was not possible to do a pooled analysis [37]. This individual study reported a reduced rate of new-onset obstructive sleep apnoea in patients undergoing bariatric surgery (new-onset

**Table 2. Outcomes for overall mortality rates relative to controls.**

| Study | Overall mortality risk | | Cardiovascular mortality risk | |
|---|---|---|---|---|
| | Hazard ratio or POR | 95% CI | Hazard ratio or POR | 95% CI |
| Arterburn [25] | 0.47 | 0.39–0.56 | — | — |
| Ceriani [36] | 0.64 | 0.29–0.97 | 0.26 | 0.09–0.72 |
| Douglas [37] | 0.97 | 0.66–1.43 | — | — |
| Eliasson [38] | 0.42 | 0.30–0.57 | 0.41 | 0.19–0.90 |
| Flum [39] | 0.67 | 0.54–0.85 | — | — |
| Kauppila [41] | 0.63 | 0.60–0.66 | 0.57 | 0.52–0.63 |
| Moussa [42] | 0.49 | 0.34–0.69 | — | — |
| Perry (age 65 years or over) [28] | 0.85 | 0.65–1.11 | — | — |
| Perry (age under 65 years) [28] | 0.72 | 0.64–0.81 | — | — |
| Pontiroli [30] | 0.52 | 0.33–0.80 | — | — |
| Singh [31] | 0.70 | 0.55–0.89 | — | — |
| Reges [32] | 0.50 | 0.40–0.61 | — | — |
| **POR** | **0.62** | **0.55–0.69** | **0.50** | **0.35–0.71** |

POR, pooled odds ratio.

obstructive sleep apnoea rate of 1.1%) compared to controls (2.0%) (HR 0.55, 95% CI 0.37–0.82, $p = 0.004$) [37].

## New-onset dyslipidaemia

Two studies reported significantly reduced incident dyslipidaemia following bariatric surgery compared to controls [32,34] (POR 0.33, 95% CI 0.14 to 0.80, $p = 0.010$) (S1 Fig). There was evidence of statistical heterogeneity (Cochran $Q = 69.5$, $p < 0.001$; $I^2 = 98.6\%$), and too few strata to assess for bias.

## New-onset ischemic heart disease

Five studies reported significantly reduced incident ischemic heart disease after bariatric surgery compared to controls [30,31,37,38,40] (POR 0.46, 95% CI 0.29 to 0.73, $p = 0.001$). There was evidence of statistical heterogeneity (Cochran $Q = 18.9$, $p < 0.001$; $I^2 = 78.8\%$) but no evidence of bias (Egger intercept = −5.62, $p = 0.119$) (S2 Fig).

## New-onset cardiac failure

Two studies reported the rate of incident cardiac failure [29,31] and found no statistically significant protective association with bariatric surgery (POR 0.23, 95% CI 0.05 to 1.10, $p = 0.066$). There was significant evidence of statistical heterogeneity (Cochran $Q = 32.5$, $p < 0.001$; $I^2 = 96.9\%$), and too few strata to assess for bias.

## Development of venous thromboembolism

Only 1 study reported incident venous thromboembolism in bariatric surgery patients relative to controls, and therefore it was not possible to undertake a pooled analysis [27]. This individual study demonstrated a reduced incidence of new-onset venous thromboembolism in bariatric surgery patients (1.7%) compared to controls (4.4%) (HR 0.60, 95% CI 0.43 to 0.84, $p = 0.003$) [27].

**Table 3. Outcomes for incident obesity-related diseases.**

| Study | T2DM | | Hypertension | | Obstructive sleep apnoea | | Dyslipidaemia | | Ischemic heart disease | | Cardiac failure | | VTE | |
|---|---|---|---|---|---|---|---|---|---|---|---|---|---|---|
| | Surgery | Control | Surgery | Control | Surgery | Control | Surgery | Control | Surgery | Control | Surgery | Control | Surgery | Control |
| Arterburn [25] | — | — | — | — | — | — | — | — | — | — | — | — | — | — |
| Backman [26] | 189 (1.0%) | 2,319 (1.3%) | — | — | — | — | — | — | — | — | — | — | — | — |
| Bailly [35] | 2,091 (2.0%) | 29,855 (13.2%) | — | — | — | — | — | — | — | — | — | — | — | — |
| Ceriani [36] | — | — | — | — | — | — | — | — | — | — | — | — | — | — |
| Douglas [37] | 158 (6.6%) | 237 (9.3%) | 79 (3.2%) | 219 (8.8%) | 36 (1.1%) | 71 (2.0%) | — | — | 40 (1.2%) | 68 (1.9%) | — | — | — | — |
| Eliasson [38] | — | — | — | — | — | — | — | — | 24 (0.4%) | 67 (1.1%) | — | — | — | — |
| Flum [39] | — | — | — | — | — | — | — | — | — | — | — | — | — | — |
| Johnson [40] | — | — | — | — | — | — | — | — | | | — | — | — | — |
| Kauppila [41] | — | — | — | — | — | — | — | — | 8 (0.3%) | 241 (1.8%) | — | — | — | — |
| Moussa [42] | — | — | — | — | — | — | — | — | — | — | — | — | — | — |
| Moussa [27] | — | — | — | — | — | — | — | — | — | — | — | — | 71 (1.7%) | 179 (4.4%) |
| Perry [28] | — | — | — | — | — | — | — | — | — | — | — | — | — | — |
| Persson [29] | — | — | — | — | — | — | — | — | — | — | — | — | — | — |
| Pontiroli [30] | 15 (9.7%) | 75 (20.8%) | 47 (30.5%) | 174 (48.3%) | — | — | — | — | — | — | 89 (0.4%) | 944 (3.7%) | — | — |
| Singh [31] | — | — | 118 (3.3%) | 567 (8.0%) | — | — | — | — | 14 (9.1%) | 52 (14.4%) | — | — | — | — |
| Reges [32] | 265 (3.2%) | 2,038 (8.1%) | 265 (3.1%) | 2038 (8.1%) | — | — | 570 (6.8%) | 3,100 (12.3%) | 49 (1.0%) | 123 (1.3%) | 19 (0.4%) | 71 (0.7%) | — | — |
| Thereaux [33] | 68 (0.4%) | 213 (1.4%) | — | — | — | — | — | — | — | — | — | — | — | — |
| Thereaux [34] | — | — | 305 (5.6%) | 760 (15.8%) | — | — | 144 (2.1%) | 565 (9.1%) | — | — | — | — | — | — |

Data are given as *n* (%).

T2DM, type 2 diabetes mellitus; VTE, venous thromboembolism.

## Comorbidity analysis by adjusted ORs

Adjusted ORs for incident comorbidities were analysed separately in order to confirm results identified during event rate analysis. These results demonstrated the same patterns identified in the event rate data, with reduced incidence of T2DM (POR 0.28, 95% CI 0.11 to 0.73, *p* = 0.009), hypertension (POR 0.32, 95% CI 0.21 to 0.47, *p* < 0.001), ischemic heart disease (POR 0.67, 95% CI 0.49 to 0.90, *p* = 0.009), and cardiac failure (POR 0.43, 95% CI 0.29 to 0.64, *p* < 0.001) in bariatric surgical patients compared to controls. These data are presented in S3 Table.

## Discussion

The results of the present study suggest that bariatric surgery is associated with reduced all-cause long-term mortality compared to appropriate control patients at a population level. This

**Fig 3. Forest plot of incident diabetes (pooled odds ratio 0.39, 95% CI 0.183 to 0.831, *p* = 0.010).**

is consistent with the findings of the Swedish Obese Subjects observational study, which demonstrated decreased overall mortality in patients with obesity who received bariatric surgery [12]. This association was also identified in previous meta-analyses that had included data from non-population-based clinical trials [22,23,44]. The current results establish that these findings are generalisable to the population at large and are not just limited to patients receiving bariatric surgery within specialised centres participating in clinical trials or publishing individual series.

The present study also suggests that bariatric surgery is associated with reduced incidence of obesity-related disease including T2DM, hypertension, and dyslipidaemia. The relative risk reductions associated with bariatric surgery were 61%, 64%, and 77% for the development of T2DM, hypertension, and dyslipidaemia, respectively. These data demonstrate the potential protective association of bariatric surgery with prevention of these obesity-related diseases. Bariatric surgery was also specifically associated with a reduction in cardiovascular mortality and development of ischemic heart disease. Bariatric surgery has previously been associated with a reduction in both macrovascular (including myocardial infarction and cerebrovascular events) and microvascular (including retinal complications, diabetic kidney disease, and peripheral neuropathy) complications relating to T2DM [10,40,45]. The reduction in cardiovascular mortality is likely accounted for by the modification of comorbidities that are known cardiovascular risk factors (T2DM, hypertension, and dyslipidaemia), as suggested in the present study.

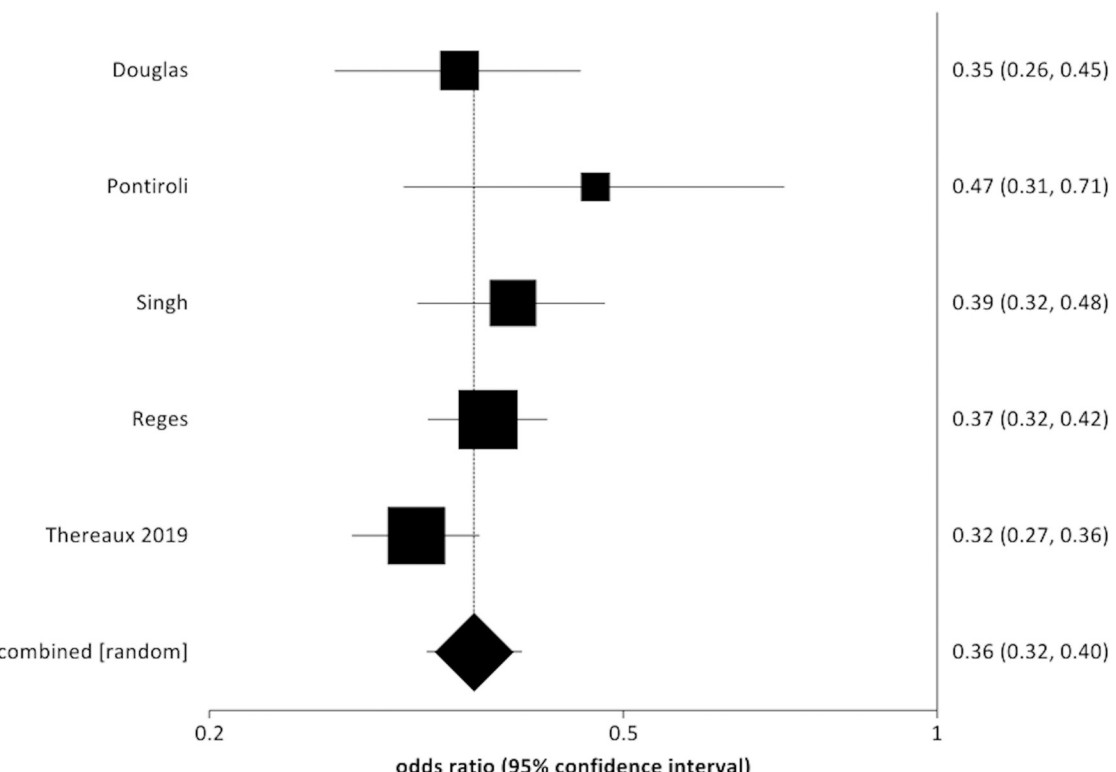

**Fig 4. Forest plot of new-onset hypertension (pooled odds ratio 0.36, 95% CI 0.32 to 0.40, $p < 0.001$).**

In addition to prevention of new-onset disease, bariatric surgery effectively improves pre-existing diseases including T2DM [10,11,46,47], dyslipidaemia [46,47], and hypertension [47]. Some of the included studies also identified improvements in remission of these diseases at a population level. Thereaux et al. [33] established that discontinuation of antidiabetic medications for patients in France with pre-existing T2DM was significantly greater in patients receiving bariatric surgery compared to controls (−49.9% versus −9.0%, $p < 0.001$) [33]. Population studies from Israel and the UK Clinical Practice Research Datalink have identified similar results [32,37]. In a separate study also included in the present analysis, Thereaux et al. [34] identified that bariatric surgery was significantly associated with discontinuation of therapy for pre-existing hypertension and dyslipidaemia relative to having obesity but not having bariatric surgery [34].

Following surgery, most patients remain in the overweight or class 1 obesity BMI category (BMI 25–35 kg/m$^2$). From data currently available, it is unclear if the overall mortality rate for patients following surgery decreases to that of the general population, decreases to that of the new equivalent BMI category, or is persistently elevated due to the period of time spent at an even higher BMI. In data from 5 Nordic countries, the standardised mortality rate (SMR) of patients receiving bariatric surgery was improved compared to non-operated patients with obesity (SMR 0.63, 95% CI 0.60 to 0.66), although mortality for these patients was still significantly increased relative to the general non-obese population (SMR 1.94, 95% CI 1.83 to 2.05) [41]. Increased BMI class has been strongly associated development of disease and increased mortality risk [48–50]; therefore, elevated mortality relative to the general population may relate to ongoing effects of existing disease before surgery. Determining the ongoing mortality

rate of patients following bariatric surgery relative to non-operated individuals within their new BMI classification remains an area for future population-based research.

Data from the UK National Bariatric Surgery Registry indicate that over 80% of patients receiving surgery have several obesity-related diseases [20]. Although most therefore have the chance to improve existing disease with bariatric surgery, our data enable us to quantify the relative risk reduction for developing disease in those disease-free at presentation. Knowledge of the relative risk reductions of 61%, 64%, and 77% for the diagnosis of incident T2DM, hypertension, and dyslipidaemia, respectively, for bariatric surgery patients relative to controls may aid the discussions between healthcare professionals and people with obesity as to whether bariatric surgery should be undertaken.

## Strengths and limitations

Our study has limitations that may affect interpretation of the results. All the studies included were large-scale, non-randomised comparative studies based upon registry data. It is therefore not possible to account for potentially important confounding variables, which may not have been equally distributed between patient groups given the non-randomised design of the studies included. One potential example is socioeconomic status, which is known to be an important predictor of all-cause mortality [51]. Previous evidence indicates that patients receiving bariatric surgery may have higher education levels and income than controls, and socioeconomic status may have influenced results [52,53]. Another example would be the higher rate of alcohol abuse disorders and schizophrenia within the control group of the study by Arterburn et al. [25] included in the present analysis. However, these large-scale studies, a strength of the paper, were selected to explore the association of bariatric surgery with long-term mortality and incidence of new obesity-related disease at a population level, with this natural variation in patient factors. The majority of studies compared results from patients receiving bariatric surgery to those of an appropriate control group, with only 1 study unable to select control group patients who also had a diagnosis of obesity [26].

An additional potential source of bias is that patients receiving bariatric surgery would have been managed within a weight loss management pathway including non-surgical treatments for obesity and close monitoring of obesity-related diseases. As the control cohorts for the studies included were taken from the general population, these individuals would likely not have been receiving any non-surgical treatment of obesity, and obesity-related diseases would likely have been less closely monitored, potentially influencing the risk of developing these diseases. It is not possible from the current dataset to evaluate separately the influence of bariatric surgery and non-surgical interventions for treatment of obesity within weight loss management programmes on rates of incident disease, and this may have led to an overestimate of the beneficial effects of bariatric surgery.

This potential source of bias can be reduced by comparing outcomes of patients receiving bariatric surgery to patients seeking treatment for obesity within a specialised medical treatment service including individual- or group-based lifestyle intervention programmes. This approach has been undertaken in a previous study by Jakobsen et al. [54] investigating long-term medical complications in a single-institution cohort of patients receiving bariatric surgery or specialised medical treatment for obesity. However, due to the population-based nature of the studies included in our analysis, it was not possible to identify which patients within the non-surgical control group may have received non-surgical treatment of obesity.

Due to heterogeneity in the definition of pre-existing disease, along with differences in the definition of remission, we did not examine disease remission rates. It was also not possible to perform a specific analysis of outcomes by procedure type in the current analysis. Our study

also did not examine specific procedure-related complications or measures of adverse outcomes associated with bariatric surgery. Bariatric surgery is known to be extremely safe, with low peri-operative mortality and complication rates [55–57]. Although other long-term potential complications of bariatric surgery exist (including gastroesophageal reflux [58], internal herniation [59], and trace element malnutrition [60]), these issues are relatively uncommon in modern bariatric surgical practice, and management strategies are available to treat these conditions should they occur. Finally, from the perspective of statistical analysis, the Egger test was utilised to measure for the presence of potential bias. This test is considered to be most reliable when greater than 10 studies are included within the analysis. Due to the limited number of studies included in some of the analyses presented here, some of these bias assessments must be interpreted carefully, although we have chosen to report these results as they still provide an indication of the presence of any potential bias.

To our knowledge, this is the first published study of pooled data from population-based studies of incident disease following bariatric surgery. Our results represent real-world data that may be generalisable to routine clinical practice. As further data accumulate, it may become clearer whether bariatric surgery reduces the incidence of new-onset obstructive sleep apnoea or venous thromboembolism. Another significant strength of the current study is that due to the nature of data collection from administrative datasets, follow-up data (such as mortality and rate of incident obesity-related disease) are externally validated outside each of the individual studies as part of the administrative database collection. This allows for completeness of data collection regarding these aspects and removes the inherent difficulties with patient follow-up data within single-institution-based studies. One method utilised to circumvent this issue within institutional studies has been to link study participants directly to data contained within administrative datasets, and this methodology was utilised successfully by Adams et al. for patients in Utah [47]. This study was not included in the present meta-analysis as all surgical patients had been collected from a single centre, although results were consistent with those identified in the current analysis (OR of incident obesity-related diseases in the surgical cohort 0.08 [95% CI 0.03 to 0.24] for T2DM, 0.23 [95% CI 0.11 to 0.49] for hypertension, and 0.12 [95% CI 0.03 to 0.46] for dyslipidaemia [47]). Results from the Swedish Obese Subjects study were also excluded from the present analysis as this study did not use a population-based design methodology, rather individuals within the control group responded to a nation-wide advert [61,62]. However, results from this study also support the findings identified here, with bariatric surgery being associated with reduced long-term mortality (HR 0.71, 95% CI 0.54 to 0.92) [12], decreased development of T2DM (OR 0.17, 95% CI 0.13 to 0.21) [63], and reduced incidence of cardiovascular events (OR 0.67, 95%CI 0.54 to 0.83) [64].

## Conclusion

This meta-analysis of large-scale registry studies indicates that patients receiving bariatric surgery have improved long-term mortality rates compared to controls at a population level. They also have significantly reduced incidence of obesity-related disease including T2DM, hypertension, dyslipidaemia, and ischemic heart disease. Healthcare providers may use the data on relative risk reduction as part of the discussion with patients considering bariatric surgery.

## Supporting information

**S1 Fig. Forest plot for development of new-onset dyslipidaemia (POR 0.33, 95% CI 0.14 to 0.80, *p* = 0.010).**
(TIF)

**S2 Fig. Forest plot of new-onset ischemic heart disease (POR 0.46, 95% CI 0.29 to 0.73, $p$ = 0.001).**
(TIF)

**S1 Table. Details of type of bariatric surgery performed within each study.** BPD, biliopancreatic diversion; DS, duodenal switch; VBG, vertical banded gastroplasty.
(DOCX)

**S2 Table. Newcastle–Ottawa Score for all included studies.**
(DOCX)

**S3 Table. Analysis of development of comorbid disease via adjusted OR data.** Data in parentheses represent 95% confidence interval. T2DM, type 2 diabetes; VTE, venous thromboembolism.
(DOCX)

**S1 Text. Additional information for literature search.**
(DOCX)

**S2 Text. PRISMA checklist.**
(DOC)

## Author Contributions

**Conceptualization:** Tom Wiggins, Sheraz R. Markar.

**Data curation:** Tom Wiggins, Nadia Guidozzi, Sheraz R. Markar.

**Formal analysis:** Tom Wiggins, Nadia Guidozzi, Sheraz R. Markar.

**Investigation:** Tom Wiggins, Richard Welbourn, Sheraz R. Markar.

**Methodology:** Tom Wiggins, Ahmed R. Ahmed, Sheraz R. Markar.

**Project administration:** Tom Wiggins, Sheraz R. Markar.

**Software:** Tom Wiggins.

**Supervision:** Richard Welbourn, Ahmed R. Ahmed, Sheraz R. Markar.

**Validation:** Richard Welbourn, Ahmed R. Ahmed, Sheraz R. Markar.

**Visualization:** Richard Welbourn, Sheraz R. Markar.

**Writing – original draft:** Tom Wiggins.

**Writing – review & editing:** Nadia Guidozzi, Richard Welbourn, Ahmed R. Ahmed, Sheraz R. Markar.

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
