## [Editor Report · Decision Letter 0]

17 Feb 2020

Dear Dr Markar, 

Thank you for submitting your manuscript entitled "Effect of bariatric surgery on all-cause mortality and incidence of obesity-related disease at a population level – systematic review and meta-analysis" for consideration by PLOS Medicine.

Your manuscript has now been evaluated by the PLOS Medicine editorial staff [as well as by an academic editor with relevant expertise] and I am writing to let you know that we would like to send your submission out for external peer review.

Kind regards,

Adya Misra, PhD,

Senior Editor

PLOS Medicine

---

## [Decision Letter · Decision Letter 1]

26 Apr 2020

Dear Dr. Markar,

Thank you very much for submitting your manuscript "Effect of bariatric surgery on all-cause mortality and incidence of obesity-related disease at a population level – systematic review and meta-analysis" (PMEDICINE-D-20-00372R1) for consideration at PLOS Medicine. 

[LINK]

In light of these reviews, I am afraid that we will not be able to accept the manuscript for publication in the journal in its current form, but we would like to consider a revised version that addresses the reviewers' and editors' comments. Obviously we cannot make any decision about publication until we have seen the revised manuscript and your response, and we plan to seek re-review by one or more of the reviewers. 

We expect to receive your revised manuscript by May 15 2020 11:59PM. Please email us (plosmedicine@plos.org) if you have any questions or concerns.

We look forward to receiving your revised manuscript. 

Sincerely,

Emma Veitch, PhD

PLOS Medicine

On behalf of:

Adya Misra, PhD

Senior Editor 

PLOS Medicine

plosmedicine.org

*Not a request but just to note that this paper was submitted, and is being considered in PLOS Medicine as part of a special issue focussed on the Determinants, Consequences and Management of Obesity

*The authors have appropriately reported the work as per PRISMA guidelines for systematic reviews, but would also be good to state (although this is not compulsory for the journal) whether the protocol for the systematic review was established and either published or registered before the start of the SR conduct - ie, either as a published protocol paper or in the PROSPERO registry (https://www.crd.york.ac.uk/PROSPERO/)

*At this stage, we ask that you include a short, non-technical Author Summary of your research to make findings accessible to a wide audience that includes both scientists and non-scientists. The Author Summary should immediately follow the Abstract in your revised manuscript. This text is subject to editorial change and should be distinct from the scientific abstract. Please see our author guidelines for more information: https://journals.plos.org/plosmedicine/s/revising-your-manuscript#loc-author-summary

*In the last sentence of the Abstract Methods and Findings section, please describe the main limitation(s) of the study's methodology. One thing to note here (see later) is that the systematic review did not, as far as this reader can tell, aim to assess adverse effects associated with bariatric surgery, only improvements on mortality and obesity-related disease.

*If possible, please change the referencing format to the PLOS Medicine style using square brackets to enclose sequential numbered callouts in the text (rather than superscript numbers) - this should be easy if referencing software was used.

*Main text covering study limitations might also include a point about not covering adverse effects in this review. 

Comments from the reviewers:

Reviewer #1: In this meta-analysis, the methods are correct, and the conclusions are sound.

There are a few details that require attention:

1. under strategy, there is no mention of paper search in PubMed

2. figures should report weight of studies

3. the analysis would be more satisfactory if reporting surgeries separately; only if no difference appears, then the surgeries can be considered together

Reviewer #2: Alex McConnachie, Statistical Review

Wiggins et al present a systematic review and meta-analysis of whole-population studies using administrative datasets, of the association between bariatric surgery and adverse outcomes, confirming the beneficial effects seen in randomised trials in general population settings. This review considers the statistical elements of the paper.

First, to be pedantic, the word "effect" in the title of the paper should perhaps be changed - since this meta-analysis is combining observational studies, the associations observed cannot be interpreted as causal effects, even if few would dispute that they are.

As a whole, the statistical methods used, their presentation, and interpretation are good. My comments are therefore relatively minor.

The literature search was reportedly carried out in January 2020. I assume that is when it was last updated?

In the final sentence of the first paragraph of the methods, the words "formal" and "formally" should maybe be removed. I am not sure what it would mean to say this was done "informally", so better to simply say it was not done.

In the final paragraph of the methods, covering the statistical methods, the word "significant" can be removed (twice). I do not think it adds anything.

For Egger's test, I have read that it is not recommended with fewer than 10 studies. This is only a rule of thumb, so the results can still be reported, but it is perhaps a limitation.

For those outcomes where there was only one study, so that a pooled analysis could not be performed, it would help the reader if the result of that one trial were shown in the text.

One phrase that crops up throughout the paper is that included studies should include an appropriate control group. It was not clear to me what was meant by "appropriate" here. Perhaps this could be expanded upon.

Also, it is stated that whilst mortality was analysed in terms of hazard ratios, the obesity related disease incidence data were analysed as event rates in each group. Did any studies present adjusted odds ratios, or use propensity scores, or other methods to control for confounding? Would it have been better to perform the meta-analysis using these adjusted ORs, rather than the raw event counts?

Reviewer #3: 

* Is the manuscript well organized and written clearly enough to be accessible to non-specialists?

Yes, this an interesting and well written manuscript.

* What are the main claims of the paper and how significant are they for the discipline?

The authors claim:

1. In the abstract (line 55-57) that "This pooled analysis has demonstrated that bariatric surgery can improve long-term all-cause mortality and reduce the incidence of obesity-related disease in patients with obesity for the whole operated population."and that "The results suggest that broader access to bariatric surgery for people with obesity may reduce the long-term sequelae of this disease and provide population-level benefits"

2. In the discussion: "This meta-analysis of large-scale registry studies indicate that patients receiving bariatric surgery have improved long-term mortality rates compared to controls at a population level."

3. In the introduction: "bariatric surgery improves all-cause mortality" (line 34 abstract), "reduces overall risk of developing multiple cancer types" (line 73), "previous pooled analyses have demonstrated that bariatric surgery reduces long-term all-cause mortality" (line86-87). These causal claims about previous research are not supported by any randomised controlled study or clinical study with an appropriate control group.

These main claims may be important for the discipline, but they are generally too optimistic by indicating causality (e.g. by the inappropriate use of the term "effect of") several places in the title, abstract, and manuscript. Importantly, all studies included in the meta-analysis were observational and none could infer causality, only associations, as mentioned in the title of several of the included studies (particularly those published in JAMA). A meta-analysis of "association studies" does not increase the evidence level to infer causation. In view of this, my advice to the authors is to remove or rephrase all undocumented causal claims in the manuscript.

* Are the claims properly placed in the context of the previous literature? Have the authors treated the literature fairly?

Yes

* Do the data and analyses fully support the claims? If not, what other evidence is required?

The claims in the first part of the discussion (lines 209-210, and 218) are partly supported by the data: "bariatric surgery is associated with reduced all-cause long-term mortality" and "reduced incidence of ", but whether the studies had "appropriate control groups" is discussable (see below).

Tthe claims of the beneficial effects (indicating causal effects) of bariatric surgery on all-cause mortality and cardiovascular mortality are overly optimistic and should be modified. Most importantly, the term "Effect" in the title "Effect of bariatric surgery.." should be removed, and may be substituted with "Associations of bariatric surgery …" as in the JAMA-papers referred to. The term "effect" should also be replaced in line 36, because the design of the current meta-analysis is not appropriate to assess any cause-effect relationship. 

Regarding other outcomes (e.g. obesity related diseases), the findings are in accordance with previously published randomised and non-randomised controlled studies. 

In my view, the main problem with the majority (if not all) of studies included in the meta-analysis is the absence of an appropriate control group, which the authors stated that they aimed to include in the methods sections (line 42 and line 112). Although this limitation is partly addressed in the discussion (lines 263-266), several other limitations with the control groups regarding imbalances in important known and unknown confounding factors in general, and in the included studies in particular, need to be addressed. First, as mentioned by the authors, less than 1% of eligible patients receive bariatric surgery. In general, and as shown in the paper by Reges et al in JAMA (ref 29), these patients probably have higher socioeconomic status (higher education and income) than controls not receiving bariatric surgery. It is well known that socioeconomic status is an important predictor of all-cause mortality. Another example is the higher prevalence of people with schizophrenia in the control group in the study by Arterburn et al. (ref 22) etc. Second, few, if any (?) control groups consisted of treatment seeking patients with obesity who received an adequate non-surgical treatment for obesity? An ethical committee would find it very difficult to accept a control group not receiving any kind of active treatment in a clinical trial, and comparing treatment-groups with no-treatment groups will probably favor the treatment group. FYI, our group has tried to partly overcome this bias by comparing treatment seeking patients with obesity who opted for either bariatric surgery or tertiary care based conservative treatment in an observational cohort study (Jakobsen GS, et al. Association of Bariatric Surgery vs Medical Obesity Treatment With Long-term Medical Complications and Obesity-Related Comorbidities. JAMA 2018;319(3):291-301) regarding remission and incidence of obesity related diseases. Third, several unmeasured confounders in the various studies might have biased the results. 

In addition, although beyond the scope of the article, the well-known detrimental long-term effects of bariatric surgery should at least be addressed in the discussion. 

Reviewer #4: Dear Editor,

Thank you for the opportunity to review the manuscript entitled: " Effects of bariatric surgery on all-cause mortality and incidence of obesity-related disease at a population level - systematic review and meta-analysis".

In the manuscript, the authors address the impact of bariatric surgery on all-cause mortality and new onset of metabolic comorbidities in a systematic review of studies published between 2000 and January, 2020.

My general impression of the study is that it is overall well conducted and the manuscript well written. The topic of the study, as well as the conclusion, are already known and accepted by bariatric surgeons and physicians with positive attitudes towards bariatric surgery. Despite the evidence supporting bariatric surgery for many patients with morbid obesity, the results still have a hard time reaching a wide acceptance. Thus, there is a need for further well written papers summarizing the beneficial effects (as well as side-effects and complications) of bariatric surgery, and these papers need to reach a wide audience of readers, such as through PLoS Medicine. Furthermore, the present study included only population-based studies, which is a novel approach.

While I have a few (mostly minor) comments and questions to the authors, it is my opinion that the manuscript should be of interest to a wide readership, and thus should be eligible for publication in PLoS Medicine.

General comments:

My main overall concern with the manuscript is related to the transparency of the presentation, and to some extent to the limitations of the studies included.

First, while all search terms are presented in the first section of the methods section, the description remains unclear in the sense that there are too may possible options for the search terms. For optimal transparency the reader should be able to redo the literature search, and end-up with similar results. I assume that the authors clustered search terms related to study group (i.e. bariatric, obesity, bariatric surgery etc.), outcome, and study level separately. If the authors could present the search-terms more specifically, it would be helpful

I fully agree with the authors that avoiding randomized trials, in particular single-center studies, can be supported by the tendency of these studies to overestimate treatment effects (and the sometimes low external validity). However, I don't fully understand why matched, prospective, multicenter studies (such as the SOS-trial) were not included in the study. Could the authors specify this decision better?

Perhaps some of these issues has been presented already in a study plan, published and registered elsewhere, but I can't find it. Was the systemic review registered before the study was conducted? Perhaps this could be specified with registration number if it was, or with a short comment if not.

Finally, it is likely that intervention groups, in general, are better controlled for comorbid disease, thus identifying more cases with disease and thereby influencing the risk for new onset of disease. Thus, there is a risk that the meta-analyses in the study overestimates the potential beneficial effect on new onset of disease. Providing remission of disease could have added to the overall view of the effect of bariatric surgery on metabolic comorbidities, but the authors have provided a reasonable explanation as to why they did not choose to include this outcome. However, a potential differential bias exists and should be presented in the limitation section.

Minor additional comments:

Abstract: Information on eligibility criteria is incomplete, and should preferably be extended in the methods section. Furthermore, a short section on the major limitations of the study could be informative.

Methods section: Did the authors contact authors from the included studies to obtain or confirm specific data from the included studies? This could have been helpful in order to have more detailed data. For instance, by contacting specific authors, surgical methods known to have inferior results (i.e. the vertical banded gastroplasty, as well as the unknown /miscellaneous group) could have been excluded. It is not a major issue though, since keeping these methods in the study is likely to underestimate the treatment effects of the intervention.

Supplementary Table 1: The studies could be more uniformly presented, preferably with author name and reference number (as raised numbers)

[LINK]

---

## [Decision Letter · Decision Letter 2]

29 May 2020

Dear Dr. Markar,

Thank you very much for re-submitting your manuscript "Association of bariatric surgery on all-cause mortality and incidence of obesity-related disease at a population level – systematic review and meta-analysis" (PMEDICINE-D-20-00372R2) for review by PLOS Medicine.

I have discussed the paper with my colleagues and the academic editor and it was also seen again by xxx reviewers. I am pleased to say that provided the remaining editorial and production issues are dealt with we are planning to accept the paper for publication in the journal.

[LINK]

We look forward to receiving the revised manuscript by Jun 05 2020 11:59PM. 

Sincerely,

Adya Misra, PhD

Senior Editor 

PLOS Medicine

plosmedicine.org

Requests from Editors:

First sentence- should “surgery” be “bariatric surgery”?

Methods and findings- please state the study designs included for clarity

Please can you provide p-values along with 95% CI as required

Conclusions must be tempered, I suggest using “suggest” instead of “demonstrated” owing to the various limitations cited in the prior section

Author summary

We need these in bullet points and using subheadings. Please see our author guidelines for more information: https://journals.plos.org/plosmedicine/s/revising-your-manuscript#loc-author-summary. We usually suggest 2-3 bullet points per subheading so please do adhere to our guidelines to avoid delays to your manuscript

Through the manuscript can you please add a space between text and the reference brackets

Introduction 

Line 91- public health need not be capitalised

Methods

Please provide the full search strategy as per PRISMA guidelines as SI files 

You may wish to briefly mention why you restricted studies between 2000 and 2020

Line 207- p values are required for up to three decimal places only 

Discussion

Please replace demonstrate with “suggest” or similar to void overreaching conclusions

Line 294-295 please rephrase “obese” to “with obesity” as you have in the rest of the submission

You may wish to rename the limitations section to “strengths and limitations”?

Table 1- please replace gender with sex 

Suppl Table 3 VTE column contains line numbers, please correct this

Please remove page numbers from the PRISMA checklist and instead use paragraphs and sections as page numbers are likely to change

Should bariatric-metabolic surgery just be ‘bariatric’?

Line 360 please avoid assertions of primacy by adding ‘to our knowledge’

Comments from Reviewers:

Reviewer #2: Alex McConnachie, Statistical Review

The authors have satisfactorily addressed each of my original observations, and I have no further comments to make.

Reviewer #3: In my opinion, the manuscript has improved considerably after revision. I have no further questions or comments.

[LINK]

---

## [Editor Report · Decision Letter 3]

22 Jun 2020

Dear Dr. Markar, 

On behalf of my colleagues and the academic editor, Dr. Ronald Ching Wan Ma, I am delighted to inform you that your manuscript entitled "Association of bariatric surgery on all-cause mortality and incidence of obesity-related disease at a population level – systematic review and meta-analysis" (PMEDICINE-D-20-00372R3) has been accepted for publication in PLOS Medicine. 

PRODUCTION PROCESS

PRESS

PROFILE INFORMATION

Thank you again for submitting the manuscript to PLOS Medicine. We look forward to publishing it. 

Best wishes, 

Adya Misra, PhD

Senior Editor 

PLOS Medicine

plosmedicine.org